# Dysregulation of T Follicular Helper and Regulatory Cells in *IRF5*-SLE Homozygous Risk Carriers and Systemic Lupus Erythematosus Patients

**DOI:** 10.3390/cells14060454

**Published:** 2025-03-19

**Authors:** Bharati Matta, Lydia Thomas, Vinay Sharma, Betsy J. Barnes

**Affiliations:** 1Center for Autoimmune Musculoskeletal and Hematopoietic Disease, The Feinstein Institutes for Medical Research, Manhasset, NY 11030, USA; bmatta1@northwell.edu (B.M.); lthomas29@northwell.edu (L.T.); 2Amity Institute of Biotechnology, Amity University Rajasthan, Jaipur 303002, India; vsharma4@jpr.amity.edu; 3Division of Pediatric Rheumatology, Cohen Children’s Medical Center, Lake Success, NY 11042, USA; 4Departments of Pediatrics and Molecular Medicine, Donald and Barbara Zucker School of Medicine at Hofstra/Northwell, Hempstead, NY 11549, USA

**Keywords:** IRF5, autoimmune, lupus, autoantibody, polymorphism, genetics

## Abstract

T follicular helper (Tfh) and T follicular regulatory cells (Tfr) are required for antibody production and are dysregulated in SLE. Genetic variants within or near interferon regulatory factor 5 (*IRF5*) are associated with SLE risk. We previously reported higher plasma cells and autoantibodies in healthy *IRF5*-SLE homozygous risk carriers. Here, we report the dysregulation of circulating Tfh and Tfr in both SLE patients and presymptomatic *IRF5*-SLE homozygous risk carriers.

## 1. Introduction

Systemic lupus erythematosus (SLE) is characterized by the production of high-affinity pathogenic autoantibodies from plasma cells (PCs) [1,2]. PCs are generated through a series of events depending upon the type of antigen. T cell-independent antigens drive PC differentiation in the absence of germinal center (GC) reaction, whereas GC is involved in PC generation in response to T cell-dependent antigens [1,2]. Both Tfh and Tfr play important roles in the GC reaction and PC differentiation [3,4,5,6]. Tfh cells are essential in T cell-dependent B cell responses in the GC and are a prerequisite for the generation of autoantibodies [7,8]. They are characterized by the expression of surface markers including chemokine C-X-C motif receptor 5 (CXCR5), inducible co-stimulator (ICOS), and programmed cell death (PD-1), as well as the intracellular marker B cell lymphoma 6 (Bcl-6). Functionally, CXCR5 is needed for GC formation, PD-1 regulates B cell selection and survival in GCs, and ICOS is required for Tfh differentiation, survival, and IL-21 production, whereas Bcl6 is required for the complete maturation of Tfh cells [7,9]. Interestingly, blood Tfh cells have been shown to be the circulating counterparts of GC Tfh cells [7,9].

In SLE, Tfh cells are shown to be essential for the GC reaction, autoantibody production, and proinflammatory cytokine production, whereas Tfr cells are required to inhibit these responses by regulating Tfh cells [3]. Notably, circulating Tfh (cTfh) cells as well as circulating Tfr (cTfr) cells were found to be positively correlated with SLE disease activity [10,11,12,13,14,15]. cTfr cells are the counterparts of GC Tfr cells and can disrupt the Tfh–B cell interaction, induce Tfh apoptosis, and thus suppress B cell functions [14].

Interferon regulatory factor 5 (IRF5) is a transcription factor previously identified as an autoimmune susceptibility gene [16,17], and genetic variants within or near *IRF5* associate strongly with SLE risk [17]. There are four primary *IRF5* genetic risk variants that comprise the homozygous SLE risk haplotype in people of European Caucasian ancestry [18]. All four genetic variants occur in the transcriptional control regions of *IRF5*; two are found in the 5′ untranslated region (UTR) (rs2004640 and a 5 bp CGGGG insertion/deletion), one in the 3′ UTR (rs10954213), and the other is 5 kb downstream of *IRF5* (rs10488631) [18]. In our previous study using this defined cohort of healthy donor *IRF5* homozygous risk carriers, we reported that they had significantly increased levels of circulating plasma cells and autoantibodies [17]. By in vivo immunizations in *Irf5*^−/−^ mice, we recently found that Irf5 plays an important role in the GC Tfh–B cell response, leading to defective PC differentiation and antibody production in a T cell-dependent manner [19]. Henceforth, we extended our immunophenotyping of healthy donor risk carriers and SLE patients to assess the contribution of *IRF5* genetic risk and/or IRF5 expression to Tfh and Tfr cell frequencies.

## 2. Methods

### 2.1. Patients

Healthy donors were genotyped by the Feinstein Genotype and Phenotype (GaP) Registry [20]. Blood was obtained from healthy donors that carry either the homozygous *IRF5*-SLE risk (n = 10) or non-risk (n = 12) haplotype consisting of rs2004640, rs10954213, rs2004640, and rs142738614/rs77571059 (male and female) [17]. As previously published [17], all genotyped donors were healthy with no personal or family history of autoimmune/inflammatory diseases or cancer. For non-genotyped healthy controls (HCs) and SLE patients, samples were randomly selected. Blood was obtained from 11 patients with active SLE (male and female) and 12 age- and sex-matched non-genotyped HC at the Northwell Health Rheumatology Clinic. Each of the patients fulfilled at least 4 of the classification criteria for SLE as defined by the American College of Rheumatology. SLE patients were evaluated for disease activity using the SLEDAI-2K that assigns individual scores to 24 descriptors; the range is from 0 to 105 [21,22]. Generally, scores ≤ 4 are considered mild disease activity, between 4 and 12 moderate disease activity, and >12 severe disease activity. Because serologic components that may not reflect disease activity are weighted equally with clinical components, each patient was also assigned a Disease Activity Score (DAS) based on clinical disease activity separate from serology [22]. A clinical SLEDAI score refers to the SLEDAI-2K score exclusive of scores for abnormal complement and/or anti-dsDNA values, as described [22]. SLE patients in the current study had a SLEDAI-2K between 4 and 12 and a DAS of 3 that constitutes moderate-to-severe disease activity.

### 2.2. Cell Isolation

Blood was collected in EDTA tubes and was diluted two times with PBS. Diluted blood was then overlayed on lymphocyte separate media (Corning, Tewksbury, MA, USA, 25-072-CV) in a 2:1 ratio. Peripheral blood mononuclear cells (PBMCs) were collected from the interphase after density gradient separation. PBMCs were either immediately stained or were frozen in 90%FBS +10%DMSO for future flow cytometry staining.

### 2.3. Flow Cytometry

PBMCs were blocked with an Fc blocker (422302, BioLegend, San Diego, CA, USA) for 10 min and then stained with antibodies against surface markers for 30 min in the dark (BioLegend: CD45-, CD3-BV711 317327, CD4-BV510 300545, CD8-PerCP 344708, CXCR5-PE 356903, PD1-APC 367405, CD69-APC-Cy7 310913, ICOS-AF700 313527; Invitrogen, Carlsbad, CA, USA: LIVE/DEAD yellow stain L34968). After staining, cells were washed 2 times in PBS +2%FBS and then fixed in 4% formaldehyde. For intracellular staining, surface-stained cells were permeabilized in 0.1% Triton X-100 for 20 min, and then stained with anti-IRF5-AF 488 (Abcam, Waltham, MA, USA, ab193245), Bcl6-PE-Cy7 (BioLegend 358511), and FoxP3-BV421 (BioLegend 320123). The samples were analyzed using BD LSRII Fortessa and FlowJo v10.1.

### 2.4. Statistics

GraphPad Prism 9 was used for statistical analysis and graphing. For comparisons, a one-way ANOVA mixed-effects, multiple-comparison analysis was performed. For correlation analysis, simple linear regression with Pearson’s correlation coefficient was used. *p* < 0.05 was considered statistically significant. Data are presented as mean ± SD.

## 3. Results

Increased IRF5 expression was observed within CD4^+^ T cells from SLE patients and healthy *IRF5*-SLE homozygous risk carriers. PBMCs from SLE patients, age- and sex-matched non-genotyped healthy controls (HCs), and healthy controls genotyped for the *IRF5*-SLE homozygous risk and non-risk haplotype were immunophenotyped by multi-color flow cytometry to assess circulating T cell subsets. We observed a significant reduction in SLE blood total CD4^+^ T cells as compared to both genotyped and non-genotyped healthy controls [23,24,25]. No differences were seen between healthy donor *IRF5*-SLE homozygous risk (R) and non-risk (NR) carriers (Figure 1A,B, Appendix A). Interestingly, we detected a significant increase in IRF5 expression within SLE CD4^+^ T cells as compared to non-genotyped healthy controls, and this difference was conserved between homozygous risk and non-risk donors [17,26,27] (Figure 1C). No significant differences were observed in CD8^+^ T cells (Figure 1A,D,E, Appendix A). We next examined T cell activation using CD69 as a marker. Somewhat surprisingly, we only detected a significant increase in the percentage of CD4^+^CD69^+^ T cells from *IRF5*-SLE homozygous risk carriers and not from SLE patients (Figure 1G, Appendix A). Lastly, there was no significant difference in IRF5 expression within activated CD4^+^CD69^+^ T cells between groups (Figure 1H), nor was there a difference in CD8^+^CD69^+^ activation levels or IRF5 expression (Figure 1I,J).

### 3.1. cTfh Cells from SLE Patients and IRF5 Risk Carriers Have Elevated IRF5 Expression

As previously reported [9,10,11,12], we found a significant increase in the percentage of CD4^+^PD1^+^CXCR5^+^ cTfh in SLE patients that are the counterpart of GC Tfh. No difference was found between genetic risk carriers (Figure 2A,B, Appendix A). Consistent with our findings in total CD4^+^ T cells, we detected significantly higher IRF5 expression in cTfh from risk as compared to non-risk carriers (Figure 2C), with no difference in Bcl6 or ICOS expression levels (Figure 2D,E). Importantly, we detected an increase in the percentage of PD1^+^ICOS^+^ cTfh cells in SLE patients, which denotes an activated state, yet differences were not seen between genotyped carriers (Figure 2F,G, Appendix A). Lastly, there was no difference in IRF5 expression within PD1^+^CXCR5^+^ICOS^+^ Tfh cells from each group (Figure 2H).

### 3.2. Higher Percentages of cTfr Cells in SLE Patients and IRF5 Risk Carriers

Given the importance of Tfr in regulatory Tfh responses, we next examined subset differences. Surprisingly, we detected a significant increase in the percentage of CD4^+^PD1^+^CXCR5^+^FoxP3^+^ cTfr cells in both SLE patients and *IRF5* risk carriers (Figure 3A,B, Appendix A). Distinct from cTfh cells, however, IRF5 expression was only significantly elevated within cTfr cells of *IRF5* risk carriers and not SLE patients (Figure 3C). We then plotted the ratio of Tfh:Tfr and found a decreased ratio in both risk compared to non-risk and SLE compared to HC that was due to increased percentages of Tfr (Figure 3D).

### 3.3. Correlation Analysis

Previously, we reported higher CD45^+^CD19^+^IgD^−^CD38^+^ circulating PCs and anti-Ro (SS-A) autoantibodies in the blood of healthy donor homozygous *IRF5* risk carriers compared to non-risk carriers [17]. As such, we used these data to perform correlation analyses with the percentages of circulating Tfh and Tfr and their IRF5 expression. Although we were unable to identify any significant correlations between the groups (Appendix A), likely due to the small sample size, we did observe the trend of positive correlation between IRF5 expression in Tfh and the percentage of cTfh, as well as with anti-Ro autoantibodies in risk carriers (Appendix A). Further, it was interesting to see a trend towards negative correlation between IRF5 expression in Tfr and anti-Ro autoantibodies (Appendix A).

## 4. Discussion

Utilizing the same cohort of SLE patients and healthy donor *IRF5*-SLE homozygous risk and non-risk carriers from the GaP registry [20], we previously reported that IRF5 expression was unchanged in B cells and myeloid cells from risk and non-risk carriers, and instead, IRF5 activation or nuclear translocation was significantly elevated in the myeloid compartment of homozygous risk carriers, which mirrored that seen in SLE patients [17]. In addition, we found increased percentages of circulating PCs and plasmacytoid dendritic cells (pDCs), elevated serum autoantibodies, increased production of interferon (IFN)α and IL-6, and increased spontaneous NETosis [17], which together categorized healthy donor risk carriers as ‘presymptomatic’ SLE. Here, we have extended the study to characterize differences in T cell subpopulations and their IRF5 expression.

T cell abnormalities have been associated with SLE [24]. Some of the phenotypic alterations are reductions in total CD4^+^ T cells and T regulatory cells, and expansion of Th17 and Th10 cells [24]. Apart from these phenotypic differences, abnormal T cell activation and cytokine production have also been reported [23,24,25]. Importantly, IRF5 has been previously shown to play an important role in the regulation of Th1 and Th17 responses, which are critical in driving SLE, and *Irf5^−/−^* mice are protected from multiple murine models of lupus, revealing defective T cell activation and global skewing towards Th2 [26,27,28,29,30,31,32]. Here, in human studies, we show that CD4^+^ T cells from *IRF5*-SLE homozygous risk carriers had higher IRF5 expression than non-risk carriers, and these cells were more activated. To date, there is little known of IRF5 expression and/or function in SLE CD4^+^ T cells or how genetic risk affects IRF5 in genotyped healthy donor CD4^+^ T cells. One study by Yan et al. showed that IRF5 expression was elevated in activated CD4^+^ T cells from healthy donors that carry a homozygous risk at rs2004640 and rs2280714, and CD4^+^ T cells from these risk carriers had increased Th1/Th17-type cytokines and reduced Th2 cytokines [27]. It is thus tempting to speculate that *IRF5* genetic risk may be an intrinsic driver of IRF5 expression within CD4^+^ T cells that mimics or initiates elevated IRF5 expression within SLE CD4^+^ T cells, contributing to their abnormalities. Further, it has been shown that SLE patients carrying the major *IRF5*-SLE homozygous risk haplotype have elevated IRF5 expression [18,26]. Interestingly, SLE patients carrying the *IRF5*-SLE risk haplotype were also found to have elevated serum IFNα levels that correlated with elevated IRF5 expression [18,33]. Prior to the current study, IRF5 expression within SLE CD4^+^ T cells had not yet been examined; however, in a previous study, we examined IRF5 activation via nuclear translocation by multispectral imaging flow cytometry and were unable to detect a difference between CD4^+^ T cells from healthy controls and SLE patients [34].

As reported by others, we found increased percentages of CD4^+^PD1^+^CXCR5^+^ Tfh cells in the circulation of SLE patients [9,10,11,12], with no apparent differences between *IRF5* risk and non-risk carriers, although IRF5 expression was significantly upregulated in cTfh from both risk carriers and SLE patients. It is thus intriguing to (re)consider the mechanism(s) by which *IRF5* homozygous risk carriers have elevated numbers of circulating PCs and autoantibodies [17], yet do not have elevated cTfh levels. We recently reported that murine Irf5 plays an important role in the ability of Tfh cells to generate GC B cells [19]. It may be that elevated IRF5 expression within cTfh determines their functional capacity rather than the proportion of circulating cells; however, more work will be required to assess the functional activity of cTfh from homozygous risk and non-risk carriers.

Somewhat distinct from the cTfh subset, we detected increased percentages of circulating Tfr cells in both SLE patients and *IRF5* risk carriers, which was unexpected since Tfr cells are known to suppress Tfh function and GC response [13,15]. However, these findings are consistent with other reports that have shown higher Tfr in SLE patients [13,14], even though there remains some controversy within the field, as others have reported reduced Tfr numbers in SLE patients [35,36]. Interestingly, it has also been reported that even though Tfr cells in SLE patients may be proportionally increased, they are functionally impaired [37]. Given the observed findings of relatively ‘normal’ cTfh levels in both genotyped and non-genotyped healthy controls, combined with the selective upregulation of IRF5 expression within cTfr of risk carriers, we surmise that risk cTfr cells are fully functional (due to elevated IRF5 expression), and are required to suppress cTfh cells in risk carriers, thus maintaining them in a presymptomatic stage of SLE. Loss of this checkpoint may be in part what flips the switch towards clinical disease.

In summary, we found dysregulated circulating Tfh and Tfr, as well as an altered Tfh/Tfr ratio, in SLE and *IRF5* homozygous risk carriers along with elevated IRF5 expression within the T cell compartment. Distinct from our previously reported findings in myeloid cells and B cells from healthy donor homozygous risk and non-risk carriers [17], the current data suggest that genetic variation in *IRF5* is a direct contributor to IRF5 expression within CD4^+^ T cells, particularly Tfh and Tfr, resulting in their dysregulation. This may be a previously undefined component of the genetic mechanism that drives presymptomatic SLE in healthy donor *IRF5* homozygous risk carriers resulting in the observed increase in circulating PCs and increased autoantibody production [17].

Importantly, while our data provide additional insight into the contribution of *IRF5* genetic risk to intrinsic T cell abnormalities seen in SLE patients, a primary limitation of this study is that we have yet to examine the functional ability of circulating Tfh and Tfr subsets from risk and non-risk carriers. Further work will be required to measure cytokine production (Th1 and Th17) from CD4^+^ T cell subsets, assess the suppressive ability of purified Tfr cells in co-culture, and address the functional capacity of Tfh to differentiate B cells into GC B cells. Another limitation is the small sample size of SLE patients with only active disease, resulting in our inability to perform rigorous correlation analyses with clinical parameters. In the future, it will also be important to address the contribution of IRF5 in SLE T cells of patients with both active and inactive disease to assess IRF5 as a functional driver of adaptive immunity in SLE.

## Figures and Tables

**Figure 1 cells-14-00454-f001:**
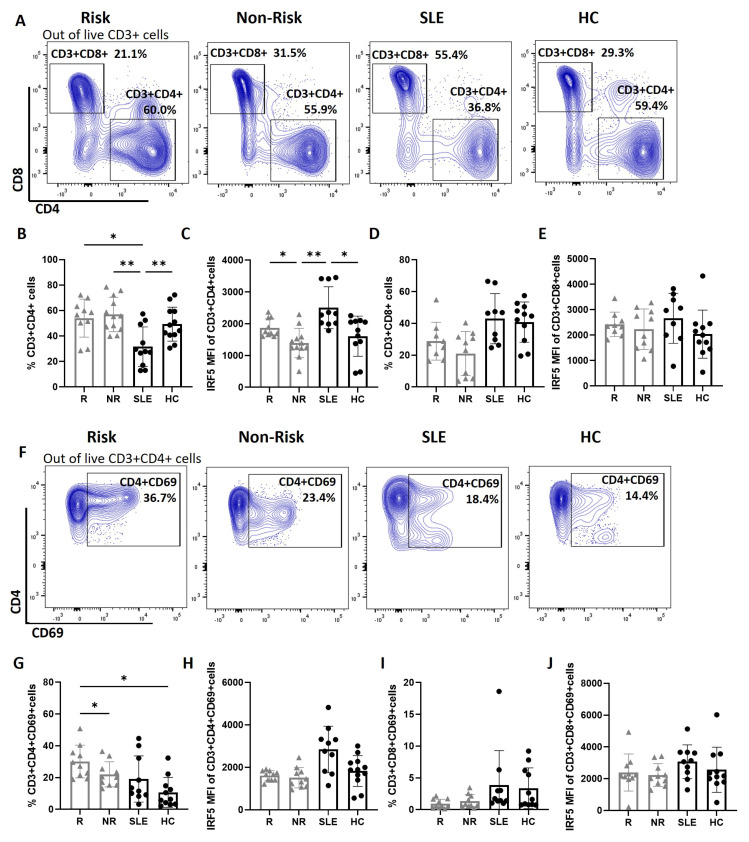
Increased IRF5 expression within CD4+ T cells. (**A**) Representative flow cytometry plots of CD3+CD4+ and CD3+CD8+ T cells from donor cohorts; *IRF5*-SLE homozygous risk carriers (R), homozygous non-risk (NR), and SLE and healthy controls (HCs). (**B**–**E**) Graphical summary of flow data showing percentages of CD3+CD4+ T cells (**B**), IRF5 MFI within CD3+CD4+ T cells (**C**), percentage of CD3+CD8+ T cells (**D**), and IRF5 MFI within CD3+CD8+ T cells (**E**). (**F**) Same as (**A**) except representative flow plots show activated CD3+CD4+CD69+ T cells. (**G**,**H**) Graphical summary of flow data showing percentages of CD3+CD4+CD69+-activated T cells (**G**), IRF5 MFI within CD3+CD4+CD69+ T cells (**H**), percentage of CD3+CD8+CD69+-activated T cells (**I**), and IRF5 MFI within CD3+CD8+CD69+ T cells (**J**). NR, n = 10–12; R, n = 9–10; SLE, n = 10–11; HC, n = 11–12. Data are presented as mean ±SD. *p* values are reported after one-way ANOVA mixed-effects analysis.**, *p* < 0.01. *, *p* < 0.05.

**Figure 2 cells-14-00454-f002:**
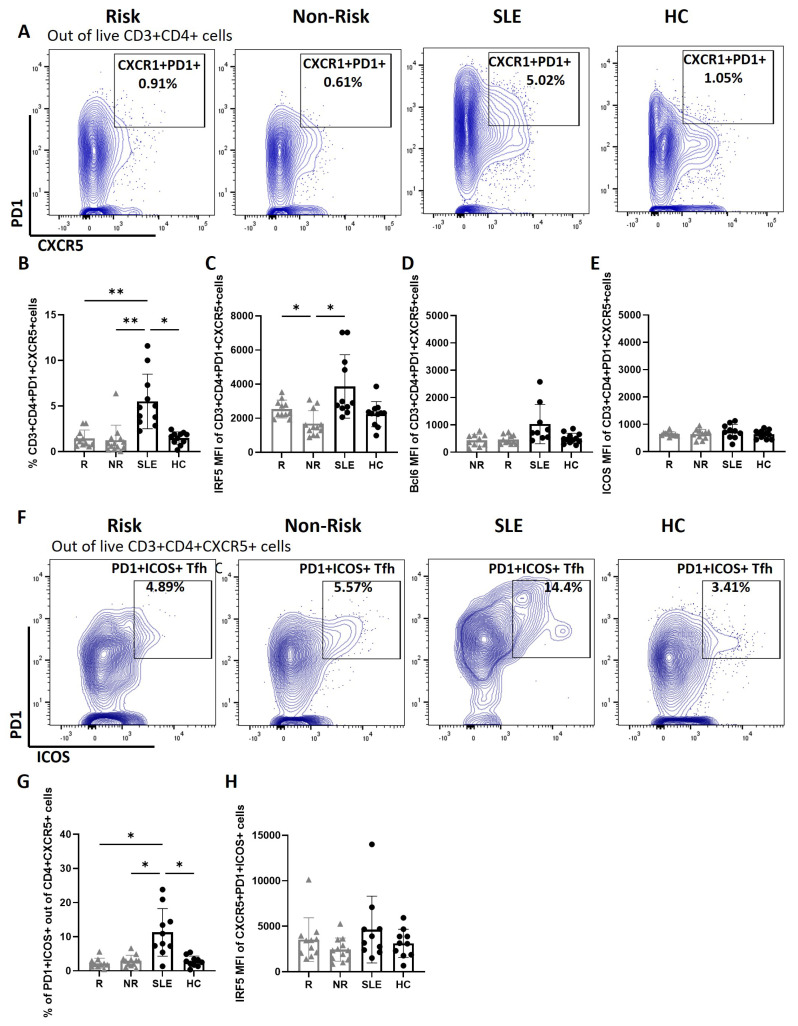
Increased circulating Tfh cells in SLE patients and *IRF5*-SLE homozygous risk carriers. (**A**) Representative flow cytometry plots of CD3+CD4+CXCR5+PD1+ Tfh cells from donor cohorts; *IRF5*-SLE homozygous risk carriers (R), homozygous non-risk (NR), and SLE and healthy controls (HCs). (**B**–**E**) Graphical summary of flow data showing percentages of CD3+CD4+CXCR5+PD1+ Tfh cells (**B**), IRF5 MFI within CD3+CD4+CXCR5+PD1+ Tfh cells (**C**), Bcl6 MFI within CD3+CD4+CXCR5+PD1+ Tfh cells (**D**), and ICOS MFI within CXCR5+PD1+ Tfh cells (**E**). (**F**) Representative flow plots of CD3+CD4+CXCR5+PD1+ICOS+ TFH cells. (**G,H**) Graphical summary of flow data showing percentage of CD3+CD4+CXCR5+PD1+ ICOS+ TFH cells (**G**) and IRF5 MFI within CD3+CD4+CXCR5+PD1+ ICOS+ TFH cells (**H**). NR, n = 10–12; R, n = 9–10; SLE, n = 10–11; HC, n = 11–12. Data are presented as mean ± SD. *p* values are reported after one-way ANOVA mixed-effects analysis. **, *p* < 0.01. *, *p* < 0.05.

**Figure 3 cells-14-00454-f003:**
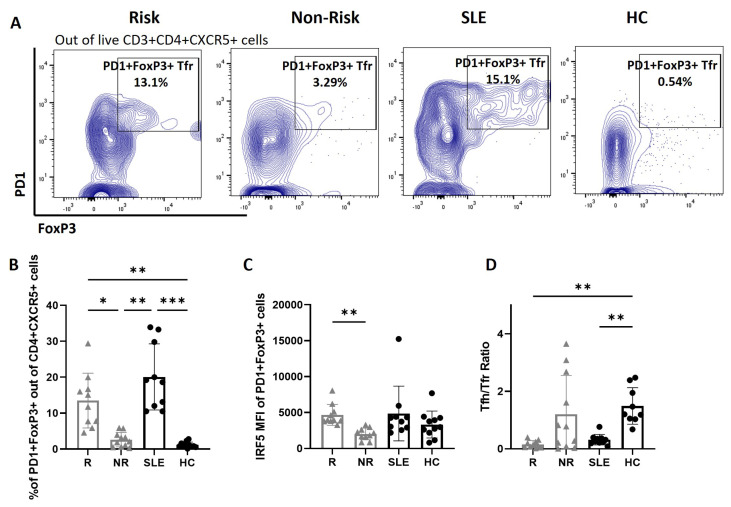
Increased circulating Tfr cells in SLE patients and *IRF5*-SLE homozygous risk carriers. (**A**) Representative flow cytometry plots of CD3+CD4+CXCR5+PD1+FOXP3+ Tfr cells from donor cohorts; *IRF5*-SLE homozygous risk carriers (R), homozygous non-risk (NR), and SLE and healthy controls (HCs). (**B**–**D**) Graphical summary of flow data showing percentages of CD3+CD4+CXCR5+PD1+FOXP3+ Tfr cells (**B**), IRF5 MFI within CD3+CD4+CXCR5+PD1+FOXP3+ TFR cells (**C**), and ratio of Tfh to Tfr cells (**D**). NR, n = 10–12; R, n = 9–10; SLE, n = 10–11; HC, n = 11–12. Data are presented as mean ± SD. *p* values are reported after one-way ANOVA mixed-effects analysis. ***, *p* < 0.001. **, *p* < 0.01. *, *p* < 0.05.

## Data Availability

All data are available in the main text of the manuscript or in the Appendix A.

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
