# Peer review of "Dysregulation of T Follicular Helper and Regulatory Cells in IRF5-SLE Homozygous Risk Carriers and Systemic Lupus Erythematosus Patients"

_cells, 2025, doi:10.3390/cells14060454_

Round 1

Reviewer 1 Report

Comments and Suggestions for Authors

I have reviewed the manuscript entitled “Dysregulation of T follicular helper and regulatory cells in IRF5-SLE homozygous risk carriers and SLE patients”, which investigates the impact of IRF5 on circulating Tfh and Tfr cells in SLE patients and healthy homozygous IRF5-LED risk carriers. While the study addresses an important topic, several issues need to be resolved before the manuscript can be considered for publication.

Major Points:

  1. Figures 1, 2, and 3 are blurry and difficult to interpret. While the authors provide descriptions, the figures themselves must be clear enough for readers to independently assess the data.
  2. Although the FMO labeling is convincing, the manuscript lacks examples of intracellular IRF5 labeling. Including this data would strengthen the conclusions.
  3. The PD1, ICOS, and FoxP3 labeling mentioned in the text of Supplemental Figure 1A is not visible in the figure. Dot plots or contour plots figures are necessary to support the authors' claims.
  4. The statement line 140 : “We then plotted ratios of Tfh:Tfr and Tfr:Tfh and found a conserved increase in Tfr from both risk carriers and SLE patients (Fig. 3D-E)” is confusing. If both figures represent similar data, why do they differ? If Figure 3E represents the Th:Tfr ratio instead, please clarify. Additionally, due to the low figure quality, it is difficult to confirm the labels.

Minor Points:

  1. It would be interesting to assess IRF5 expression in other T cell subpopulations, such as Th1 and Th17 cells.
  2. Increasing the sample size would improve the statistical power and robustness of the study’s findings.
  3. The manuscript would benefit from correlation analyses between IRF5 expression levels and relevant clinical parameters to strengthen its translational impact.

In conclusion, while the manuscript addresses an interesting topic, the above concerns should be addressed to ensure clarity.

Author Response

Reviewer 1:

I have reviewed the manuscript entitled “Dysregulation of T follicular helper and regulatory cells in IRF5-SLE homozygous risk carriers and SLE patients”, which investigates the impact of IRF5 on circulating Tfh and Tfr cells in SLE patients and healthy homozygous IRF5-LED risk carriers. While the study addresses an important topic, several issues need to be resolved before the manuscript can be considered for publication.

Major Points:

1. Figures 1, 2, and 3 are blurry and difficult to interpret. While the authors provide descriptions, the figures themselves must be clear enough for readers to independently assess the data.

Although the figures were generated as high-resolution tiffs, we think that insertion of them into the word .doc reduced their resolution after pdf generation. We apologize for this. In the revision, we have modified to further enhance visibility and are also able to submit the high-resolution tiffs independent of the manuscript.

2. Although the FMO labeling is convincing, the manuscript lacks examples of intracellular IRF5 labeling. Including this data would strengthen the conclusions.

Thank you for the suggestion, we have now added this in Supplemental Figure 1C.

3. The PD1, ICOS, and FoxP3 labeling mentioned in the text of Supplemental Figure 1A is not visible in the figure. Dot plots or contour plots figures are necessary to support the authors' claims.

We have now added more clear labels and contour plots to Supplemental Figure 1A.

4. The statement line 140 : “We then plotted ratios of Tfh:Tfr and Tfr:Tfh and found a conserved increase in Tfr from both risk carriers and SLE patients (Fig. 3D-E)” is confusing. If both figures represent similar data, why do they differ? If Figure 3E represents the Th:Tfr ratio instead, please clarify. Additionally, due to the low figure quality, it is difficult to confirm the labels.

We understand that it was confusing and have now presented only one ratio (Tfh:Tfr) to further simplify. Even though, importantly, both of the original ratios were generated from the same data; they would just be opposite each other.

Minor Points:

1. It would be interesting to assess IRF5 expression in other T cell subpopulations, such as Th1 and Th17 cells.

Yes, this is certainly of interest and will need to be done in a larger patient cohort. We have included aspects of this in the Discussion. 

2. Increasing the sample size would improve the statistical power and robustness of the study’s findings.

We absolutely agree. The GaP registry at the Feinstein Institutes continues to genotype new healthy donors and we are in the process of identifying additional carriers of the IRF5-SLE risk and non-risk haplotype for phenotyping and functional analysis. We have included this as a limitation of the study.

3. The manuscript would benefit from correlation analyses between IRF5 expression levels and relevant clinical parameters to strengthen its translational impact.

We performed correlation analyses for GaP donors but were unable to perform a rigorous analysis in SLE patients due to the small sample size and limited heterogeneity in disease activity. We have included this limitation in the Discussion.

Reviewer 2 Report

Comments and Suggestions for Authors

The authors present an interesting study assessing the influence of IRF5 on Tfh and Tfr frequencies. The study is well conducted, just some minor comments. 

  • Why did you use the former classification criteria to select SLE patients? Moreover reference on Class criteria should be reported.
  • Figures are blurry and impossible to read. 

Author Response

Reviewer 2:

The authors present an interesting study assessing the influence of IRF5 on Tfh and Tfr frequencies. The study is well conducted, just some minor comments. 

  • Why did you use the former classification criteria to select SLE patients? Moreover reference on Class criteria should be reported.

We apologize for not clarifying the SLE patient classification and have now included the details in the Methods section for Patients.

  • Figures are blurry and impossible to read. 

We apologize for the figures being blurred and difficult to read. We have modified for clarity and are now able to upload as tiffs. 

Reviewer 3 Report

Comments and Suggestions for Authors

The authors found dysregulated circulating Tfh, Tfr and Tfh/Tfr ratios in SLE
and IRF5 homozygous risk carriers along with elevated IRF5 expression within the T cell
compartment. 

The authors should discuss the relationship of interferon regulatory factor 5 gene polymporphism with systemic lupus erythematosus susceptibility.  The authors should discuss within their discussion the relationship of inerferon regulatory factor 5 gene polymorphism in healthy individuals and SLE patients with their findings.

The authors should discuss the contribution of their findings to the understanding of SLE pathogenesis in both healthy individuals and carriers of interferon regulatory factor 5 polymporphisms. 

Author Response

Reviewer 3:

The authors found dysregulated circulating Tfh, Tfr and Tfh/Tfr ratios in SLE
and IRF5 homozygous risk carriers along with elevated IRF5 expression within the T cell compartment. 

The authors should discuss the relationship of interferon regulatory factor 5 gene polymporphism with systemic lupus erythematosus susceptibility. The authors should discuss within their discussion the relationship of inerferon regulatory factor 5 gene polymorphism in healthy individuals and SLE patients with their findings.

The authors should discuss the contribution of their findings to the understanding of SLE pathogenesis in both healthy individuals and carriers of interferon regulatory factor 5 polymporphisms. 

We thank the Reviewer for their suggestions and now include all of these aspects in the Discussion.

Round 2

Reviewer 1 Report

Comments and Suggestions for Authors

Congratulations on this work

Reviewer 3 Report

Comments and Suggestions for Authors

The authors have made the necessary amendments. The manuscript may now be published.